# The Relationship between Ambient Fine Particulate Matter (PM$_{2.5}$) Pollution and Depression: An Analysis of Data from 185 Countries

Ravi Philip Rajkumar

Department of Psychiatry, Jawaharlal Institute of Postgraduate Medical Education and Research (JIPMER), Puducherry 605 006, India; jd0422@jipmer.ac.in; Tel.: +91-413-229-6280

**Abstract:** Several studies have identified a relationship between air pollution and depression, particularly in relation to fine particulate matter (PM$_{2.5}$) exposure. However, the strength of this association appears to be moderated by variables such as age, gender, genetic vulnerability, physical activity, and climatic conditions, and has not been assessed at a cross-national level to date. Moreover, certain studies in this field have yielded negative results, and there are discrepancies between the results obtained in high-income countries and those from low- and middle-income countries. The current study examines cross-sectional and longitudinal associations between the incidence of depression in each country, based on Global Burden of Disease Study data, and the average national level of PM$_{2.5}$ based on the World Health Organization's database, over the past decade (2010–2019). The observed associations were adjusted for age, gender, level of physical activity, income, education, population density, climate, and type of depression. It was observed that while PM$_{2.5}$ levels showed significant cross-sectional associations with the incidence of depression, longitudinal analyses were not suggestive of a direct causal relationship. These findings are discussed in the light of recent contradictory results in this field, and the need to consider the intermediate roles of a number of individual and environmental factors.

**Keywords:** air pollution; PM$_{2.5}$; depression; inflammation; ecological analysis; climate; gender





## 1. Introduction

Depression is the commonest mental disorder worldwide. Based on data from the Global Burden of Disease Study (2019), it is estimated that over 279 million people suffer from depression, and depression is the 13th leading cause of disease burden and disability at a global level [1]. The pathophysiology of depression is complex, and involves an interaction between genetic risk, mediated by multiple genes of small effect, and a wide range of environmental factors [2,3]. Much of the research on gene–environment interactions in depression has been focused on the harmful effects of psychological or social stress [4,5]. However, there is a rapidly expanding evidence base suggesting that various aspects of an individual's living environment, such as access to green areas and exposure to various forms of pollution, are also significantly associated with depression [6]. Among these environmental factors, air pollution has been associated with depression most frequently and consistently. Air pollution, particularly exposure to fine particulate matter (PM$_{2.5}$) and nitrogen oxides (NO$_x$), has been associated with depression in youth [7], adults in the general population [8,9], women in the post-partum period [10,11], and the elderly [12]. Exposure to these forms of air pollution has also been associated with worse outcomes in patients with depression, such as suicide [13] and acute episodes or exacerbations of depression requiring contact with emergency health services [14].

There are several pathophysiological processes that may account for the increased rates of depression observed after exposure to various air pollutants. Exposure to PM$_{2.5}$ has been associated with increased expression of genes involved in the immune inflammatory

response [15], which is known to be associated with disease onset and severity and treatment outcomes in depression [16,17]. In animal models, $PM_{2.5}$ exposure led to activation of microglia, leading to central nervous system inflammation and neuronal toxicity, which was associated with depressive-like behaviors [18]. These effects were associated with elevated release of tumor necrosis factor-alpha (TNF-$\alpha$), which is also elevated in patients suffering from depression [19]. Specific air pollutants may also interact with innate genetic vulnerability factors to increase susceptibility to depression. These gene x environment interactions appear to lead to increased systemic inflammation and oxidative stress, thereby altering the functioning of specific brain circuits involved in memory, problem solving, and the ability to cope with stress [20]. Thus, air pollution may act synergistically with chronic psychological or social stress in the genesis of depression [21]. Over and above these brain-specific mechanisms, particulate matter exposure is associated with an increased risk of several chronic medical conditions, including cardiovascular, cerebrovascular, and respiratory illnesses. These chronic illnesses are themselves associated with an increased risk of depression [22].

However, two recent meta-analyses of the existing literature have reached divergent conclusions. In the first analysis, published in 2020, depression was associated with short-term exposure to nitrogen dioxide ($NO_2$), but not with other forms of air pollution [23]. In the second analysis, published in 2022, depression was significantly associated with short-term and long-term exposure to $PM_{2.5}$ as well as with exposure to nitrogen dioxide, sulfur dioxide, ozone, and carbon monoxide [24]. Similarly, some studies of the relationship between air pollution and depression in specific countries have yielded negative results [25]. A part of this variation may be due to the specific type of pollutant(s) ($PM_{2.5}$, $PM_{10}$, nitrogen oxides, sulfur dioxide, or ozone) included in the analysis, but much of the variability in these results can be ascribed to host-related factors. These factors include age, gender, vitamin D status, level of physical activity, income, education, population density, and physical health, as well as innate genetic variations [10–12,15,20,26–28]. Climatic conditions can also influence the impact of air pollution [29]. Finally, cross-sectional or retrospective associations are insufficient to establish a clear causal relationship between exposure to air pollution and depression. It is therefore possible that a more accurate picture of the relationship between air pollution and depression could be obtained if

(a)   At least some of these interacting or confounding factors were taken into account;
(b)   Individual pollutants were examined individually;
(c)   A longitudinal study design was adopted;
(d)   Data from low- and middle-income countries, where air pollution is often above acceptable levels, were included.

The research presented in this paper represents an initial step in this process. In this study, data on the levels of a specific pollutant (fine particulate matter with a diameter less than 2.5 microns, $PM_{2.5}$) are examined in relation to the incidence of depression across 185 countries and territories, while attempting to account for the confounding effect of the variables enumerated above, over the period 2010–2019. Based on the results of earlier researchers, the following hypotheses were formulated:

**Hypothesis 1 (H1).** *Fine particulate matter pollution ($PM_{2.5}$) is positively correlated with the incidence of depression across multiple countries and territories.*

**Hypothesis 2 (H2).** *The correlation between $PM_{2.5}$ and depression will remain significant after adjusting for the effects of age, gender, physical activity, income, and geographical location.*

**Hypothesis 3 (H3).** *Longitudinal methods of analysis will demonstrate the possibility of a causal relationship between $PM_{2.5}$ exposure and depression.*

## 2. Materials and Methods

The current study is an ecological association study, using both cross-sectional and longitudinal methods of analysis, based on the most recently available data for 185 countries and regions. For the purpose of this study, atmospheric levels of fine particulate matter of diameter less than 2.5 microns (henceforth designated $PM_{2.5}$) were selected as the exposure of interest, and the estimated incidence of depression for each country was taken as the dependent variable. $PM_{2.5}$ was selected as the exposure for the following reasons:

- Across research studies, $PM_{2.5}$ has been one of the pollutants most consistently associated with the risk of depression [7,9,11,12,25].
- There are biologically plausible mechanisms, identified through translational and clinical research, linking $PM_{2.5}$ exposure with depression [15,18,20].
- Reliable and recent cross-national data on levels of $PM_{2.5}$ are available from a reliable source for the years 2010 and 2018 [30].

### 2.1. Data Sources

Information on the estimated levels of $PM_{2.5}$ for each country was obtained from the World Health Organization's Global Health Observatory (GHO). The GHO provides estimates of this parameter for a total of 185 countries and regions, and these values are available for the years 2010 and 2018. Data on average annual $PM_{2.5}$ levels for each country for both of these years were used as the chief dependent variable in this study [30].

Depression is a chronic condition with an episodic course; therefore, examining associations between $PM_{2.5}$ and the prevalence of this disorder could yield inaccurate results due to the confounding effects of the natural history of depression. Moreover, there is evidence that short-term $PM_{2.5}$ exposure can trigger the onset of new episodes of depression [14]. Therefore, data on the incidence of depression (defined as the number of new cases of depression per unit population, expressed as a percentage) were retrieved through specific queries from the Global Burden of Disease Study database for the years 2010 and 2019 [31]. This database provides the most recent, nation-wide estimates of the incidence, prevalence, and disease burden associated with specific illnesses, based on the most recent epidemiological research [1,32]. Given the exploratory nature of the current study, data on the incidence of depression were retrieved (i) for the general population as a whole, and (ii) for men and women separately. The confounding effects of age were adjusted for by obtaining age-standardized estimates of the incidence of depression from the database. These estimates are adjusted for the age distribution of a "standard" population, and thereby control for variations in age distribution across countries.

Apart from age and gender, which were adjusted for in the manner described above, several other factors may influence the association between $PM_{2.5}$ exposure and depression. These confounding variables, the rationale for their inclusion, and the data source for each of these variables are summarized in Table 1 below.

**Table 1.** Confounding factors analyzed in the current study.

| Variable | Rationale for Inclusion | Data Source |
|---|---|---|
| Gross national income (GNI) per capita | Income levels may mediate the link between air pollution and mental health [26] | World Bank database [33] |
| Average years of education per adult | Within income groups, education may influence the link between air pollution and depression [26] | Our World in Data [34] |
| Prevalence of insufficient physical activity in adults (%) | Physical activity may moderate the impact of air pollution on depression [27] | WHO Global Health Observatory [35] |
| Population density | Population density may influence the links between air pollution and mental health [26] | World Population Review [36] |
| Distance from the equator (absolute value of the latitude) | Climatic conditions can influence the impact of air pollution, and may also influence vitamin D levels [29,37] | Google Earth [38] |

*2.2. Data Analysis*

All study variables were tested for normality prior to analysis using the Shapiro–Wilk test. The estimated incidence of depression conformed to an approximately Gaussian distribution. The other study variables deviated significantly from this distribution ($p < 0.01$, Shapiro–Wilk test) and were subjected to a natural logarithmic transformation prior to data analysis.

In the first phase of the data analysis, unadjusted bivariate correlations (Pearson's $r$) were computed for the associations between the concentration of $PM_{2.5}$ and the incidence of depressive disorder for both the years (2010 and 2019) for which data were available. To rule out the possibility of a Type I error involving $\mathbf{H_1}$, Bonferroni's correction was applied to the correlation matrix, and both corrected and uncorrected $p$-values were mentioned in the relevant section of the Results. The magnitude of observed correlations was classified as poor ($0.1 < r < 0.3$), fair ($0.3 \leq r < 0.6$), moderate ($0.6 \leq r < 0.8$), or strong ($r \geq 0.8$) based on standard guideline values for biomedical research [39]. These analyses were then repeated with adjustments for the confounding variables listed in Table 1, using Pearson's partial $r$.

Next, the possibility of a threshold effect was also examined. According to the World Health Organization guidelines, a 24 h mean exposure level of 15 µg/m$^3$ is considered "safe" for fine particulate matter ($PM_{2.5}$) [40]. To examine this possibility, countries were classified as "safe" or "unsafe" based on an estimated $PM_{2.5}$ of $\leq$15 µg/m$^3$ and >15 µg/m$^3$, respectively, and the incidence of depression was compared across these two groups using the independent samples $t$-test. As a confirmatory analysis, correlations between $PM_{2.5}$ and the incidence of depression were computed separately for each group.

Finally, the possibility of a causal relationship between average $PM_{2.5}$ levels and the incidence of depression was assessed using two longitudinal models: cross-lagged regression analysis and repeated measures analysis of covariance (RM-ANCOVA). In the first model, cross-correlations between $PM_{2.5}$ levels and the incidence of depression for the years 2010 and 2019 were computed: a causal relationship could be inferred only if the correlation between $PM_{2.5}$ (in 2010) and depression (in 2019) was significantly stronger than the correlation between depression (in 2010) and $PM_{2.5}$ (in 2019) [41]. The second model involved two steps: first, a general linear model (repeated measures analysis of variance, RM-ANOVA) was carried out to see if the incidence of depression was significantly different across both time points. If this was the case, then $PM_{2.5}$ levels were introduced as a covariate to test for an interaction effect.

All statistical tests were two-tailed, and a significance level of $p < 0.05$ was used for all bivariate analyses.

## 3. Results

Data were available for a total of 185 countries and regions. The mean estimated incidence of depression was $0.75 \pm 0.22\%$ in 2010, and $0.74 \pm 0.21\%$ in 2019. When incidence was examined by gender, estimates were $0.58 \pm 0.19\%$ (2010) and $0.57 \pm 0.18\%$ (2019) for men, and $0.92 \pm 0.28\%$ (2010) and $0.91 \pm 0.26\%$ (2019) for women.

The median (inter-quartile range) estimated concentration of $PM_{2.5}$ for all the countries included in this study was 20.25 (17.73) µg/m$^3$ in 2010, and 22.01 (24.63) µg/m$^3$ in 2019.

Longitudinal comparisons of these variables revealed that the incidence of depression decreased in the period 2010–2019 (paired-samples $t = -2.58$, $p = 0.011$), but the magnitude of this decrease was small (Cohen's $d = 0.19$). In contrast, average $PM_{2.5}$ levels increased modestly during this period (Wilcoxon's signed-rank test statistic = 5404.5, $p < 0.001$, effect size = 0.37).

*3.1. Bivariate Correlations between $PM_{2.5}$ and the Incidence of Depression*

The results of unadjusted and adjusted bivariate correlations between $PM_{2.5}$ and the incidence of depression for the years 2010 and 2019 are presented in Table 2. In the unadjusted analyses (Table 2, Row 1), it can be observed that average $PM_{2.5}$ levels were positively correlated with the occurrence of new episodes of depression, independent of

gender. These associations remained significant after Bonferroni's correction, but were poor to fair in strength. A visual inspection of the scatter plots suggests that the relationship between these variables was not entirely linear, with a slight reduction in the incidence of depression observed at higher $PM_{2.5}$ levels. At lower levels of atmospheric $PM_{2.5}$, there appears to be a linear or monotonic relationship; however, at levels of $PM_{2.5}$ above 25–30 $\mu g/m^3$ or so, this association was not observed (Figure 1).

**Table 2.** Bivariate correlations between average annual $PM_{2.5}$ levels and the incidence of depression across countries.

| Variable | Correlation with $PM_{2.5}$ Levels, 2010 | Correlation with $PM_{2.5}$ Levels, 2019 |
|---|---|---|
| Depression, incidence (unadjusted) | | |
| Total | **0.28 (<0.001)** | **0.37 (<0.001)** |
| Male | **0.27 (<0.001)** | **0.40 (<0.001)** |
| Female | 0.27 (<0.001) | **0.32 (<0.001)** |
| Depression, incidence (adjusted) * | | |
| Total | 0.21 (0.011) | **0.33 (<0.001)** |
| Male | 0.21 (0.011) | **0.34 (<0.001)** |
| Female | 0.19 (0.020) | **0.29 (<0.001)** |

* Adjusted for gross national income, average adult education, population density, percentage of adults engaging in sufficient physical activity, and distance from the equator. Values marked in **bold** indicate statistically significant findings after Bonferroni's correction.

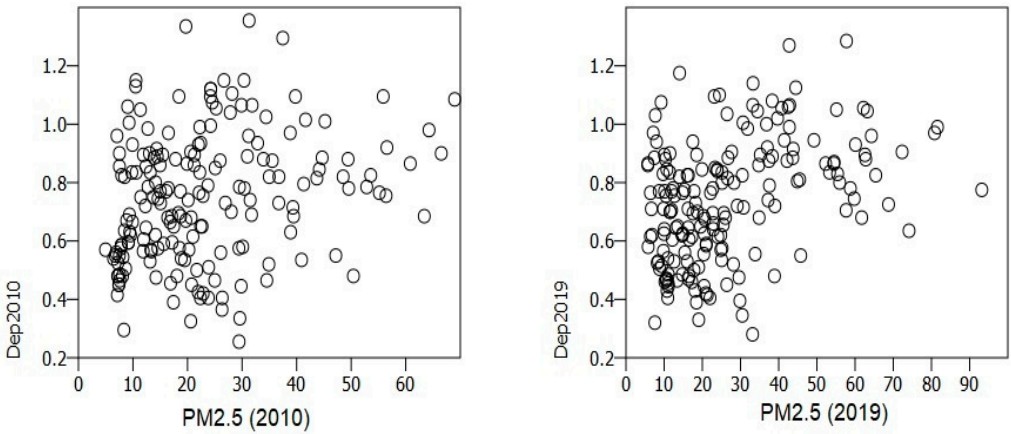

**Figure 1.** Scatter plots of the relationship between average annual $PM_{2.5}$ and the incidence of depression in 2010 and 2019.

In the adjusted analyses (Table 2, Row 1), $PM_{2.5}$ levels remained significantly and positively correlated with the incidence of depression after adjusting for national income, population density, education, physical activity, and distance from the equator. The magnitude of these correlations was only slightly altered by these adjustments. However, when Bonferroni's correction was applied, these findings remained significant for the year 2019, but not for 2010. Therefore, these results provide partial support for $H_1$ at lower levels of $PM_{2.5}$, but only inconsistent support for $H_2$. Correlations between these confounding variables themselves and the incidence of depression are provided in Supplementary Table S1. In these supplementary analyses, only the level of adult education was significantly and negatively correlated with the incidence of depression.

All values are given as Pearson's *r* (*p*-value). Results highlighted in **bold** indicate statistically significant findings after applying Bonferroni's correction for a 5 × 5 table.

### *3.2. Evaluation of a Possible Threshold Effect*

To examine whether the World Health Organization's guideline cut-off for $PM_{2.5}$ exposure (15 µg/m$^3$) was significantly associated with depression, countries were classified as being above or below this threshold for each year of interest. In the year 2010, 120 countries were above the safety threshold; in 2019, 127 countries were above this threshold. Following this, the bivariate analyses reported in Table 2 were repeated for each sub-group for both years, and these results are presented in Table 3. It can be noted that in the year 2010, the association between $PM_{2.5}$ and the incidence of depression was stronger in those countries below the guideline value for safety, though it was significant in both groups. On the other hand, in the year 2019, the association between $PM_{2.5}$ and depression was stronger in countries above the guideline value, and was not significant in the remaining countries. In this context, it should be noted that in countries above the threshold, $PM_{2.5}$ levels increased significantly over the study period, from a mean of 28.5 to 34.7 µg/m$^3$ ($t = 6.30$, $p < 0.001$), while in countries below the threshold, $PM_{2.5}$ levels decreased modestly but significantly in the same period, from a mean of 11.6 to 10.4 µg/m$^3$ ($t = -3.47$, $p = 0.001$).

**Table 3.** Bivariate correlations between average annual $PM_{2.5}$ and the incidence of depression, grouped according to WHO safety guidelines.

| Variable | Correlation with $PM_{2.5}$ Levels, 2010 | Correlation with $PM_{2.5}$ Levels, 2019 |
|---|---|---|
| Depression, incidence (2010) | | |
| Total | 0.22 (0.015) | **0.38 (0.002)** |
| Male | 0.21 (0.025) | **0.39 (0.001)** |
| Female | 0.21 (0.019) | **0.35 (0.005)** |
| Depression, incidence (2019) | | |
| Total | **0.48 (<0.001)** | −0.13 (0.318) |
| Male | **0.48 (<0.001)** | −0.16 (0.238) |
| Female | **0.43 (<0.001)** | −0.11 (0.400) |

All values are given as Pearson's *r* (*p*-value). Results highlighted in **bold** indicate statistically significant findings after applying Bonferroni's correction for a 5 × 5 table.

To extend these results from a longitudinal perspective, countries were classified according to whether or not their average annual $PM_{2.5}$ levels had increased beyond the guideline values in the period 2010–2019. This phenomenon was observed in twenty countries; however, these countries did not differ significantly from the others in terms of the incidence of depression (independent-sample $t = 0.26$, $p = 0.797$). Moreover, the correlation between $PM_{2.5}$ levels and depression in this subset of countries was not statistically significant ($r = 0.21$, $p = 0.379$). These inconsistent results suggest that there is no clear threshold effect for a link between $PM_{2.5}$ and depression.

### *3.3. Longitudinal Associations between $PM_{2.5}$ and the Incidence of Depression*
3.3.1. Cross-Lagged Regression

The results of the cross-lagged regression analysis, both unadjusted and adjusted, are presented in Table 4. In these analyses, it can be observed that none of the cross-lagged regression coefficients were statistically significant. These results suggest that **H₃** should be rejected: in other words, there is no significant evidence for a causal relationship between $PM_{2.5}$ levels and depression in the study period.

**Table 4.** Cross-lagged regression analyses between $PM_{2.5}$ levels and the incidence of depression for the period 2010–2019.

| Variable | Correlation with $PM_{2.5}$ Levels, 2010 | Correlation with $PM_{2.5}$ Levels, 2019 | Cross-Lagged Regression Coefficient | Significance Level |
|---|---|---|---|---|
| Depression, incidence (%) | | | | |
| Total | 0.29 (<0.001) | 0.34 (<0.001) | −0.051 | 0.491 |
| Male | 0.28 (<0.001) | 0.37 (<0.001) | −0.061 | 0.409 |
| Female | 0.27 (<0.001) | 0.30 (<0.001) | −0.018 | 0.807 |
| Depression, incidence (%) * | | | | |
| Total | 0.23 (0.006) | 0.30 (<0.001) | −0.068 | 0.357 |
| Male | 0.23 (0.005) | 0.31 (<0.001) | −0.049 | 0.507 |
| Female | 0.21 (0.013) | 0.26 (0.001) | −0.055 | 0.457 |

* Adjusted for gross national income, average adult education, population density, percentage of adults engaging in sufficient physical activity, and distance from the equator.

### 3.3.2. General Linear Model

As a second test of the possibility of a causal relationship between $PM_{2.5}$ and depression, repeated measures analyses of covariance were carried out for the incidence of depression while taking the baseline (2010) $PM_{2.5}$ level as a covariate. The results of these analyses are presented in Table 5. In these analyses, it can be observed that there was a slight but statistically significant decrease in the incidence of depression irrespective of gender. However, there was no significant interaction effect when $PM_{2.5}$ levels were introduced as a covariate. These results also suggest that $H_3$ should be rejected.

**Table 5.** Repeated measures analyses of covariance of the relationship between $PM_{2.5}$ and longitudinal changes in the incidence of depression.

| Variable | Test Statistic (F) | Significance Level |
|---|---|---|
| Depression, total | 6.66 | 0.011 |
| Depression, total $\times$ $PM_{2.5}$ | 0.26 | 0.610 |
| Depression, male | 7.76 | 0.006 |
| Depression, male $\times$ $PM_{2.5}$ | 0.03 | 0.986 |
| Depression, female | 4.95 | 0.027 |
| Depression, female $\times$ $PM_{2.5}$ | 0.49 | 0.483 |

### 3.4. Additional Analyses

In light of the negative results obtained in Section 3.3, the bivariate correlation between the percentage of change in $PM_{2.5}$ levels and the percentage of change in the incidence of depression between 2010 and 2019 was computed. This association was found to be statistically insignificant ($r = 0.10$, $p = 0.197$), and remained insignificant when adjusting for possible confounding variables (partial $r = 0.03$, $p = 0.718$).

### 4. Discussion

The results of the current study suggest that the association between atmospheric $PM_{2.5}$ levels and the incidence of depression is significant cross-sectionally, but not longitudinally. In this study, the use of age-standardized incidence estimates and the inclusion of separate analyses for men and women were used to correct for the possible were of age and gender, and data from two time points ten years apart (2010 and 2019) was used to examine longitudinal associations and possible causality. The cross-sectional association between $PM_{2.5}$ and depression remained significant in both genders, but retained its significance only partially after adjustment for the confounding variables identified by earlier researchers. In other words, this study was able to confirm $H_1$, but was unable to provide full support for $H_2$. In contrast, all the three methods used to assess longitudinal relationships and possible causality did not yield significant results; as a result, the study's findings do not provide any consistent support for $H_3$.

Examination of the data for a possible threshold effect did not yield consistent findings. Associations between $PM_{2.5}$ levels and depression appeared to be stronger in countries with levels above the WHO safety threshold in 2010, but the reverse was observed in 2019. These results are difficult to interpret in terms of linear causation. The figures in the first row of Table 3 are consistent with earlier studies suggesting an effect of prolonged $PM_{2.5}$ exposure on depression, with correlation coefficients increasing in strength over time; however, this explanation does not account for the figures in the second row, in which the correlation coefficients actually decreased in magnitude and significance at the second time point. It is of course possible that this could reflect a decrease in $PM_{2.5}$ levels in the latter group of countries, many of which belong to the high-income category and may have implemented more efficient pollution control measures. This possibility is supported by the fact that mean levels of $PM_{2.5}$ increased in the first group of countries over the study period, but decreased in the second group. Though this finding requires replication, it is potentially valuable, as it suggests that efforts to reduce atmospheric $PM_{2.5}$ pollution may reduce the likelihood of depression over time.

These results suggest that the association between atmospheric $PM_{2.5}$ levels and the incidence of depression, though statistically significant, is not of a direct causal nature. These results appear to be at variance with the large number of studies that suggest that such a causal relationship exists [7–14]. However, a closer look at the evidence suggests that the link between air pollution and depression is complex and influenced by several confounding factors. Meta-analyses of the available literature have yielded conflicting results. For example, a meta-analysis of all available studies up to the year 2019 failed to find a significant association between either short- or long-term exposure to $PM_{2.5}$ and depression [23]. A subsequent meta-analysis published in 2022 estimated that the odds ratio for the relationship between $PM_{2.5}$ and depression was 1.009 for short-term exposure and 1.074 for long-term exposure, suggesting modest effects [24]. Both these reviews highlighted the risk of publication bias in the existing literature, suggesting that negative results might not have received appropriate exposure.

More recent studies have underlined the complex and inconsistent nature of this relationship. For example, a study of the links between short-term $PM_{2.5}$ exposure and depression suggested that this risk may be significantly elevated in specific populations, such as women and the elderly [42]. The results of a cohort study of elderly American adults reported a significant relationship between long-term $PM_{2.5}$ exposure and depression, underling the possible age-specific nature of this relationship [43]. A retrospective analysis of data on over 100,000 adults from the United Kingdom suggested that $PM_{2.5}$ exposure was modestly associated with a nonspecific increase in the risk of mental disorders, and not specifically depression; it was also observed that this risk was likely to be mediated by genetic factors [44]. A French study with a similar sample size found that the association between various air pollutants (including $PM_{2.5}$) and depressive symptoms was strongest in adults with a personal history of social and economic disadvantage, characterized by a lower income, lower educational attainment, and living in deprived regions [45]. A study of 15,000 middle-aged and elderly Chinese adults found that the association between ambient $PM_{2.5}$ and depression was not significant when taking into account the effects of alcohol consumption [46]. These findings suggest that a multitude of genetic and environmental factors can influence the relationship between $PM_{2.5}$ and depression.

Two recent studies from China highlighted the key role played by psychological factors in the relationship between $PM_{2.5}$ exposure and depression. In the first, subjective satisfaction with air quality was significantly associated with depressive symptoms in young and middle-aged adults, and this relationship was further influenced by individuals' perception of their own health [47]. In the second, individual methods of coping with stress and trauma were found to influence the link between high levels of air pollution and depression [48]. The above results suggest that individual perceptions of, and reactions to, air quality may be as important as objective measures of air pollution in determining psychological outcomes such as depression. These findings are consistent with the cognitive

theory of depression, in which negative cognitions related to the self and the external world or environment can lead to a depressed mood and associated behavioral changes [49].

It is also important to note that external air pollution is not the only likely source of pollution at play, especially in low- and middle- income countries. Several studies from these countries have highlighted the role of indoor air pollution, derived from cooking fuels, in triggering or worsening depression [50–52]. Likewise, in countries with few legal restrictions on tobacco use in public places, second-hand exposure to cigarette smoke may also be associated with depressive symptoms [53,54]. As data on the exact magnitude of these variables are difficult to collect at a cross-national level, it was not possible to incorporate them into the analysis. However, it remains plausible that the overall level of exposure to air pollution, both indoor and outdoor, may correlate more strongly with depression than measures of outdoor air pollution alone; this is supported by a recent report suggesting that controlling for indoor pollution weakens the apparent association between outdoor air pollutants and depression [55].

Another factor that could have contributed to the lack of a direct longitudinal association in this study is the difficulty in accounting for measures to reduce atmospheric pollution and their efficacy, as suggested above. While short-term exposure to $PM_{2.5}$ could trigger episodes of depression, long-term effects due to prolonged exposure could be attenuated by effective pollution control strategies. The current study was not designed to address this specific hypothesis, but the comparisons of mean $PM_{2.5}$ changes for the groups of countries in Table 3 are consistent with it.

The negative results in the current study could also be due to specific methodological issues which have been emphasized by recent research in this field. For example, levels of $PM_{2.5}$ may vary significantly across a country, resulting in significant associations in some regions but not in others [56]. Second, the definition of the depressive phenotype has varied significantly across studies. Some researchers have measured depressive symptoms [10,45,46] which may or may not be clinically significant and which can be transiently experienced by many individuals, while others have used a more stringent definition based on syndromic criteria for depression [12–14,44]. As the data used in this study were based on the latter definition, this could account for some of the discrepancies in the reported associations. Third, phenomena such as air pollution do not exist in isolation. For example, in urban settings, air pollution may be accompanied by reduced green space availability, excessive exposure to artificial light, and noise pollution, which may independently contribute to the risk of depression [57,58]. Fourth, other climatic variables, such as ambient temperature and sunlight exposure, may influence the links between particulate matter exposure and depression [59–61]. Finally, apart from the psychological and social factors listed above, individual lifestyle variables such as diet and exercise can either enhance or attenuate the links between exposure to pollutants and depression [46,62].

It is also worth noting that most existing research on air pollution and depression is based on data from industrialized, high-income countries, with few studies from low- and middle-countries [9]. In the current dataset, there was a significant negative correlation between national income and $PM_{2.5}$ levels which appeared to increase over time ($r = -0.24$, $p < 0.001$ in 2010; $r = -0.47$, $p < 0.001$ in 2019). This is consistent with existing data which suggest that the effects of air pollution may be more severe in lower-income countries, particularly where measures to reduce pollution are often absent or ineffective [63,64]. In contrast, in countries with low levels of outdoor air pollution and effective legislation to minimize this phenomenon, the associations between $PM_{2.5}$ and depression may be weak or absent [25,65].

Overall, these results of the current study are consistent with the hypothesis advanced by Hu et al., which states that the relationship between air pollution and mental health outcomes, such as depression, is heterogeneous and influenced by a wide range of confounding factors [26]. This is illustrated in Figure 2, which highlights the large number of intermediate variables that can influence the relationship between a specific type of pollutant, such as $PM_{2.5}$, and depression. It is worth noting that many of these factors can

interact with each other. For example, an objective deterioration in air quality can lead to a subjective dissatisfaction with the environment, or ideas of helplessness concerning an individual's capacity to change this situation, and this could lead to depression. Likewise, both air pollution and economic disadvantage could lead to a reduction in social capital, such as reduced contact between neighbors, and this could increase vulnerability to depression. Such interactions could be modeled in studies of individual subjects in the community, allowing for a better delineation of the relative contributions of each risk or protective factor to depression.

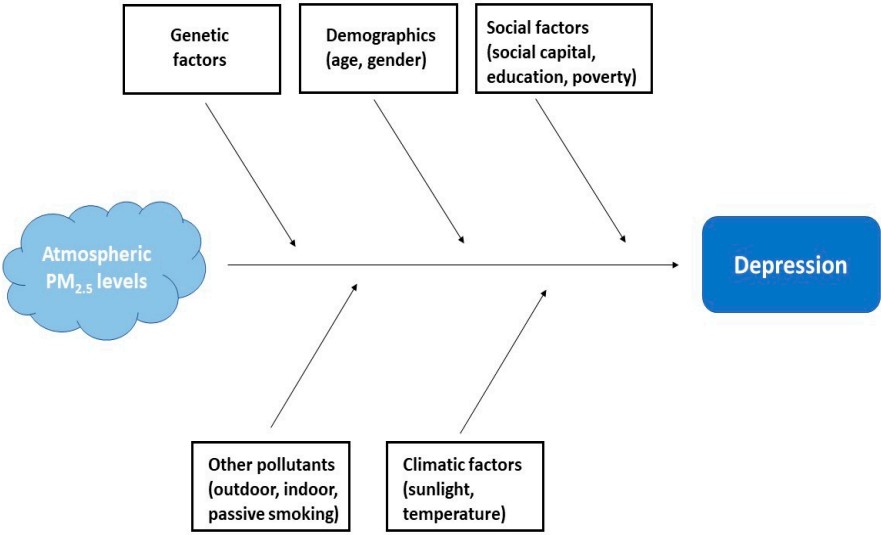

**Figure 2.** The complex nature of the relationship between PM$_{2.5}$ and depression.

There are certain limitations of the current study which must be borne in mind when interpreting its results. First, they are based on country-level estimates, and cannot be directly extrapolated to individuals [66]. Second, there is a certain degree of uncertainty regarding the reliability of the estimates for the incidence of depression, particularly in lower-income countries [67]. Third, certain other confounding factors may also influence the link between PM$_{2.5}$ and depression, as illustrated in Figure 2, but these could not be adjusted for in the current study due to a lack of cross-national data. Fourth, the association between PM$_{2.5}$ and depression may not be specific; instead, prolonged PM$_{2.5}$ exposure may be a non-specific risk factor for a number of mental illnesses, including anxiety disorders, bipolar disorder, and psychosis [68]. Fifth, only annual averages of PM$_{2.5}$ levels for each country were available. Though such data have been used in earlier studies [69], their use is limited by the fact that more fine-grained analyses of weekly or monthly changes could not be carried out [70]. Additionally, as longitudinal data were available only for the period 2010–2019, it was not possible to examine the possible causal role of exposure to PM$_{2.5}$ over a longer period of time. Sixth, this study examined PM$_{2.5}$ levels because data were available on this variable; it was not possible to examine the effect of other relevant pollutants, such as nitrogen oxides and sulfur dioxide [23,24]. Seventh, due to a lack of relevant data, it was not possible to examine the effects of pollution control measures in each country over the study period. Finally, it was not possible to analyze the effects of relevant genetic variants on vulnerability to depression following PM$_{2.5}$ exposure, as relevant data on allele frequencies are available only for a small number of countries.

## 5. Conclusions

Despite certain limitations and divergences from the existing evidence, the results presented in this paper are of importance, not just for healthcare professionals, but for all those involved in mitigating the health and environmental impacts of outdoor air pollution. The key results obtained from the current data suggest that

- There is a cross-sectional, positive correlation between $PM_{2.5}$ levels and the incidence of depression, but this relationship is of a linear or monotonic nature only to a limited extent, and was weakened after correcting for possible confounders;
- The strength of the association between $PM_{2.5}$ and depression increased over time in countries with a $PM_{2.5}$ level above the WHO-recommended threshold of 15 $\mu g/m^3$; these countries also experienced a significant increase in $PM_{2.5}$ levels over the study period;
- On the other hand, in countries with a $PM_{2.5}$ level below this threshold, the relationship between $PM_{2.5}$ and depression weakened over time; these countries showed a slight but significant reduction in $PM_{2.5}$ over the study period;
- Linear methods of analysis did not reveal a clear-cut longitudinal relationship between $PM_{2.5}$ levels; however, these results should be interpreted in light of the preceding three findings.

The discrepancies between the cross-sectional and longitudinal results in this study suggest the need for a more critical and in-depth analysis of the links between $PM_{2.5}$ exposure and depression, as has been pointed out by other researchers in this field [26,71,72]. The study findings also suggest that reductions in $PM_{2.5}$ levels over time may reduce the strength of the association between $PM_{2.5}$ exposure and depression in some cases. Given the personal, social, and economic burden associated with depression, particularly in low- and middle-income countries where patients with this disorder are often undiagnosed and untreated, prompt measures to prevent or at least mitigate the possible effects of $PM_{2.5}$ on this disorder are required. Such measures would require effective changes in policy and a commitment to change by individuals, industries, and other non-state actors. In addition, measures should be taken to enhance resilience to the possible "depressogenic" effects of $PM_{2.5}$ exposure: these could include individual-level strategies to enhance coping skills and stress management, attempts to improve educational achievement and to reduce economic deprivation and inequality within communities, and measures to reduce exposure to other inhaled toxins, such as second-hand smoke or indoor pollutants from cooking fuels.

**Supplementary Materials:** The following supporting information can be downloaded at: https://www.mdpi.com/article/10.3390/atmos14030597/s1, Table S1: Bivariate correlations between the incidence of depression and potential confounding factors.

**Funding:** This research received no external funding.

**Institutional Review Board Statement:** Not applicable.

**Informed Consent Statement:** Not applicable.

**Data Availability Statement:** All data used for the purpose of this study are in the public domain and have been referenced in the paper. A complete data sheet is available from the author on request.

**Conflicts of Interest:** The author declares no conflict of interest.

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
