# Peer review of "The Relationship between Ambient Fine Particulate Matter (PM2.5) Pollution and Depression: An Analysis of Data from 185 Countries"

_atmosphere, doi:10.3390/atmos14030597_

Round 1

Reviewer 1 Report (Previous Reviewer 1)

Manuscript Number; Atmosphere-2263279

Title; The relationship between ambient fine particulate matter (PM2.5) pollution and depression: an analysis of data from 185 countries

Evaluation; Minor revision

The study aims to demonstrate the relationship between PM2.5 and assess the impact of depression in many countries. All of the main text, typing errors and subscripts (PM2.5) should be checked. 

Author Response

The study aims to demonstrate the relationship between PM2.5 and assess the impact of depression in many countries. All of the main text, typing errors and subscripts (PM2.5) should be checked. 

Response: I thank the reviewer for highlighting this important issue. All typographical errors and errors in subscripts have been corrected to the best of my knowledge in the revised manuscript. Missing footnotes for the tables have also been added.

Reviewer 2 Report (Previous Reviewer 2)

1. Line 12-14: it is a contradictory statement. Recent publications showed that middle and low income countries are highest stake holders in publications related to air pollution and health. Kindly update the facts.

2. As evident by Table 2 and Figure 1, there is no correlation between PM and depression in 2010 and 2019. So what is the base of hypothesis then?

3. Table 3 is really saying the results and I appreciate it. I suggest to highlight these results in a better way in the revised manuscript.

4. Kindly make bullet points in the conclusion and improve the section with the findings stated in the discussion section.

Author Response

1. Line 12-14: it is a contradictory statement. Recent publications showed that middle and low income countries are highest stake holders in publications related to air pollution and health. Kindly update the facts.

Response: I apologize for this error. It has been corrected in the revised manuscript as follows:

Abstract (lines 13-14): "Moreover, certain studies in this field have yielded negative results, and there are discrepancies between the results obtained in high-income countries and those from low- and middle-income countries."

Discussion (lines 444-445): "It is also worth noting that most studies showing a direct dose-response relationship between PM2.5 and depression are based on data from industrialized, high-income countries, with few such results from low- and middle-countries [9]." It has been clarified that the difference is in the study results and not in terms of stakeholders.

2. As evident by Table 2 and Figure 1, there is no correlation between PM and depression in 2010 and 2019. So what is the base of hypothesis then?

Response: I agree with the reviewer's comment and have revised the text to make this clearer as follows:

Lines 207-215: "A visual inspection of the scatter plots suggests that the relationship between these variables was not entirely linear, with a slight reduction in the incidence of depression observed at higher PM2.5 levels. At lower levels of atmospheric PM2.5, there appears to be a linear or monotonic relationship; however, at levels of PM2.5 above 25-30 µg/m3 or so, this association was not observed (Figure 1). This non-linear relationship was supported by the results of attempts at curve fitting, in which a logarithmic relationship between PM2.5 and depression explained more of the variance in depression (adjusted R2 = 10%) than a linear relationship (adjusted R2 = 2.8%)."

Lines 224-225: "Therefore, these results provide partial support for H1 at lower levels of PM2.5"

The same has also been mentioned in the Discussion (lines ): "These results suggest that the association between atmospheric PM2.5 levels and the incidence of depression is not of a consistently causal nature across the different countries studied, and may be non-linear (Figure 1)."

The association between percentage changes in PM2.5 and depression over the study period has also been included in the revised manuscript in lines 306-315:

"In the light of the negative results obtained in section 3.3, the bivariate correlation between the percentage of change in PM2.5 levels and the percentage of change in the incidence of depression between 2010-2019 was computed. There was a positive, but weak correlation between these variables (r = .22, p = .002), suggesting that increases in PM2.5 were positively correlated with increases in depression; however, this association was not significant when adjusting for potential confounders (partial r = .108, p = .196). When this analysis was conducted based on sub-groups, a positive correlation between increases in PM2.5 and depression was observed in countries whose PM2.5 levels were above the WHO safety threshold (r = .20, p = .022) but not in those below this level (r = .08, p = .537)."

This point has also been highlighted in the Conclusions (lines):

"...there is a cross-sectional, positive correlation between PM2.5 levels and the incidence of depression, but this relationship is of a linear or monotonic nature only to a limited extent, and was weakened after correcting for possible confounders; curve fitting suggests that this relationship may be logarithmic rather than linear."

3. Table 3 is really saying the results and I appreciate it. I suggest to highlight these results in a better way in the revised manuscript.

Response: I thank the reviewer for their kind comments. In line with their suggestion, the following has been added to the revised manuscript:

Lines 247-251 (results): "In this context, it should be noted that in countries above the threshold, PM2.5 levels increased significantly over the study period, from a mean of 28.5 to 34.7 µg/m3 (t = 6.30, p < .001), while in countries below the threshold, PM2.5 levels decreased modestly but significantly in the same period, from a mean of 11.6 to 10.4 µg/m3 (t = -3.47, p = .001)."

Lines 268-270 have been deleted in view of the reviewer's suggestion, and new findings have been presented in Tables 5 and 6 and in lines 275-277

"In view of the differences observed between countries above and below the WHO safety threshold, presented in section 3.1, cross-lagged regression analyses were also carried out for both groups of countries separately. These results are presented in Table 5."

as well as lines 281-289:

"The results of these analyses differ significantly from those of the sample as a whole. In countries whose PM2.5 levels are within the WHO threshold of 15 µg/m3, a significant causal relationship can be considered: decreases in ambient PM2.5 levels were associated with a significant decrease in the incidence of depression over time. On the other hand, in countries above this guideline value, the reverse appears to hold true: baseline levels of depression appear to predict subsequent increases in PM2.5 over the study period. Thus, these results can be considered to partly support H3 in the subgroup of countries with (a) a baseline PM2.5 below 15 µg/m3 and (b) a significant decrease in PM2.5 levels over the study period, but not in other countries."

and lines 296-300:

"There was no significant interaction effect when PM2.5 levels were introduced as a covariate; however, there was a significant interaction effect when the WHO safety threshold of 15 µg/m3 was included in the model, particularly in the case of men. These results suggest that H3 should be rejected for the overall dataset, but may still be supported in the sub-group of countries described above."

Lines 338-354 (New paragraph added to discussion): "Examination of the data for a possible threshold effect did not yield consistent findings. Associations between PM2.5 levels and depression appeared to be stronger in countries with levels above the WHO safety threshold in 2010, but the reverse was observed in 2019. The figures in the first row of Table 3 are consistent with earlier studies suggesting an effect of prolonged PM2.5 exposure on depression, with correlation coefficients increasing in strength over time. However, this explanation does not account for the figures in the second row, in which the correlation coefficients actually decreased in magnitude and significance at the second time point. It is of course possible that this could reflect a decrease in PM2.5 levels in the latter group of countries, many of which belong to the high-income category and may have implemented more efficient pollution control measures. This possibility is supported by the fact that mean levels of PM2.5 increased in the first group of countries over the study period, but decreased in the second group, and receives further support from the longitudinal findings in Table 5 and Table 6. Though this finding requires replication, it is potentially valuable, as it suggests that efforts to reduce atmospheric PM2.5 pollution may reduce the likelihood of depression over time. On the other hand, at pollution levels above a certain threshold, there may not be a dose-response relationship between PM2.5 levels and depression."

Lines 418-423 (added to Discussion): 

"Another factor that could have contributed to the lack of a direct longitudinal association in this study is the difficulty in accounting for measures to reduce atmospheric pollution and their efficacy, as suggested above. While short-term exposure to PM2.5 could trigger episodes of depression, long-term effects due to prolonged exposure could be attenuated by effective pollution control strategies. The current study was not designed to address this specific hypothesis, but the comparisons of mean PM2.5 changes for the groups of countries in Table 3 are consistent with it."

4. Kindly make bullet points in the conclusion and improve the section with the findings stated in the discussion section.

Response: I thank the reviewer for this valuable suggestion. The Conclusion has been rewritten with bullet points and updated findings as follows (lines 497-537):

"Despite certain limitations and divergences from the existing evidence, the results presented in this paper are of importance, not just for healthcare professionals, but for all those involved in mitigating the health and environmental impacts of outdoor air pollution. The key results obtained from the current data suggest that:

  • there is a cross-sectional, positive correlation between PM5 levels and the incidence of depression, but this relationship is of a linear or monotonic nature only to a limited extent, and was weakened after correcting for possible confounders; curve fitting suggests that this relationship may be logarithmic rather than linear;
  • the strength of the association between PM5 and depression increased over time in countries with a PM2.5 level above the WHO-recommended threshold of 15 µg/m3; these countries also experienced a significant increase in PM2.5 levels over the study period;
  • on the other hand, in countries with a PM5 level below this threshold, the relationship between PM2.5 and depression weakened over time; these countries showed a slight but significant reduction in PM2.5 over the study period;
  • linear methods of analysis did not reveal a clear-cut longitudinal relationship between PM5 levels and depression; however, a causal relationship could be demonstrated in countries with a baseline PM2.5 level below 15 µg/m3; these countries experienced a decrease in PM2.5 levels over the study period.

The discrepancies between the cross-sectional and longitudinal results in this study, and the mixed findings in the latter, suggest the need for a more critical and in-depth analysis of the links between PM2.5 exposure and depression, as has been pointed out by other researchers in this field [26, 71, 72]. The study findings also suggest that reductions in PM2.5 levels over time may reduce the strength of the association between PM2.5 exposure and depression in some countries, and that above a certain level of PM2.5 exposure, there is a constant but not dose-dependent increase in the incidence of depression. Given the personal, social and economic burden associated with depression, particularly in low- and middle-income countries where patients with this disorder are often undiagnosed and treated, prompt measures to prevent or at least mitigate the possible effects of PM2.5 on this disorder are required. Such measures would require effective changes in policy and a commitment to change by individuals, industries and other non-state actors, with a goal of adhering as closely as possible to the WHO safety guidelines. In addition, measures should be taken to enhance resilience to the possible “depressogenic” effects of PM2.5 exposure: these could include individual-level strategies to enhance coping skills and stress management, attempts to improve educational achievement and to reduce economic deprivation and inequality within communities, and measures to reduce exposure to other inhaled toxins, such as second-hand smoke or indoor pollutants from cooking fuels."

This manuscript is a resubmission of an earlier submission. The following is a list of the peer review reports and author responses from that submission.

Round 1

Reviewer 1 Report

Manuscript Number; Atmosphere-2096043

Title; The relationship between ambient fine particulate matter (PM2.5) pollution and depression: an analysis of data from 185 countries

Evaluation; Rejected

The study aims to demonstrate the relationship between PM2.5 and assess the

impact of depression in many countries.

          Generally, the paper is interesting because the presented results are related to air pollution in the whole picture on a global scale However, there are many similar studies worldwide providing information on local air quality and using standard methodology and analysis.

          The main contribution of the presented results is related to the statistical data and obtained database. My major concern is related to the scientific novelty - the originality is the weak part of this paper and the authors need to better address what new science is presented in this paper.

Technical suggestions:

Typing errors and subscripts (PM2.5) should be checked.

Reviewer 2 Report

1. How use of mental health services is a worst outcome? Does the author mean to say that if one is using mental health service, it is the worst thing? Line 44, Introduction.

2. So the data for PM2.5 the author has analysed is annual average of a particular country for the year 2018? Please state it this way.

3. Now if I understood the methodology, the results are drawn based on correlations between depression (different categories) and the annual average concentrations of PM2.5 over a specific country.  The author used his knowledge of statistics and performed different kind of correlations to show confidence in results.  Do the author think that stating the reason for depression based on annual average values of a particular year could be concentration of PM2.5, which is annual average? The higher/lower annual average can be due to some extreme pollution event and definitely due to some specific season. On the other hand, developing depression is not instantaneous either. The patients might be having reasons for depressions, varying from years to months. How the author ruled out these possibilities? I suggest if author want to see the correlation only (an no other analysis) as a indicator of pollution impact on health, the author should deal with daily or weekly data for both pollution and patient statistics. The whole idea of correlating the yearly average values of 2018 is just showing the things in a coincidental way, not in a scientific way. I am quoting here Neil deGrasse Tyson, a famous American astrophysicist: "But to measure cause and effect... you must ensure that a simple correlation, however tempting it may be, is not mistaken for a cause. In the 1990s the stork population of Germany increased and the German at-home birth rate rose as well. Shall we credit storks for airlifting the babies?"